# Exploratory adaptation in large random networks

Hallel I. Schreier[1,2], Yoav Soen[3] & Naama Brenner[1,4]

The capacity of cells and organisms to respond to challenging conditions in a repeatable manner is limited by a finite repertoire of pre-evolved adaptive responses. Beyond this capacity, cells can use exploratory dynamics to cope with a much broader array of conditions. However, the process of adaptation by exploratory dynamics within the lifetime of a cell is not well understood. Here we demonstrate the feasibility of exploratory adaptation in a high-dimensional network model of gene regulation. Exploration is initiated by failure to comply with a constraint and is implemented by random sampling of network configurations. It ceases if and when the network reaches a stable state satisfying the constraint. We find that successful convergence (adaptation) in high dimensions requires outgoing network hubs and is enhanced by their auto-regulation. The ability of these empirically validated features of gene regulatory networks to support exploratory adaptation without fine-tuning, makes it plausible for biological implementation.

[1] Network Biology Research Laboratories, Technion—Israel Institute of Technology, Haifa 32000, Israel. [2] Interdisciplinary Program for Applied Mathematics, Technion—Israel Institute of Technology, Haifa 32000, Israel. [3] Department of Biomolecular Sciences, Weizmann Institute of Science, Rehovot 76100, Israel. [4] Department of Chemical Engineering, Technion—Israel Institute of Technology, Haifa 32000, Israel. Correspondence and requests for materials should be addressed to N.B. (email: nbrenner@technion.ac.il).

The ability to organize a large number of interacting processes into persistently viable states in a dynamic environment is a striking property of cells and organisms. Many frequently encountered perturbations (temperature, osmotic pressure, starvation and more), trigger reproducible adaptive responses[1–3]. These were assimilated into the organism by variation and selection over evolutionary time. Despite the large number and flexible nature of these responses, they span a finite repertoire of actions and cannot address all possible scenarios of novel conditions. Indeed, cells may encounter severe, unforeseen situations within their lifetime, for which no effective response is available. To survive such challenges, a different type of ad-hoc response can be employed, utilizing exploratory dynamics[4–8].

The capacity to withstand unforeseen conditions was recently demonstrated and studied using dedicated experimental models of novel challenge in yeast[9–12] and flies[13]. Adaptive responses exposed in these experiments involved transient changes in the expression of hundreds of genes, followed by convergence to altered patterns of expression. Analysis of repeated experiments showed that a large fraction of the transcriptional response can vary substantially across replicate trajectories of adaptation[10,12]. These findings suggest that coping with unforeseen challenges within one or a few generations relies on induction of exploratory changes in gene regulation over time in an individual[5,6].

Several properties of gene regulatory networks may support such exploratory adaptation. These include a large number of potential interactions between genes[14], context-dependent plasticity of interactions[15–18] and multiplicity of microscopic configurations consistent with a given phenotype[19]. Despite these properties, the feasibility of acquiring adaptive phenotypes by random exploration within a single organism remains speculative and poorly understood. In particular, it is not known how exploration may converge rapidly enough in the high dimensional space of possible configurations? what determines the efficiency of this exploration? and what ensures the stabilization of new phenotpes?

Here we address these open questions by introducing a network model of gene regulation, which demonstrates the capacity to adapt by exploratory dynamics in a single cell (as opposed to selection on existing variation in a population). Exploration is triggered by failure to satisfy a newly-imposed external demand, and is implemented by a random walk in the space of network configurations. Exploration relaxes if and when the system reaches a stable state satisfying this demand. We show that the success of this exploratory adaptation in high dimension requires that the network include outgoing hubs. Adaptive capability is further enhanced by autregulation of these outgoing hubs. Since these are both well-known properties of gene regulatory networks, our findings establish a basis for a biologically plausible mode of adaptation by exploratory dynamics.

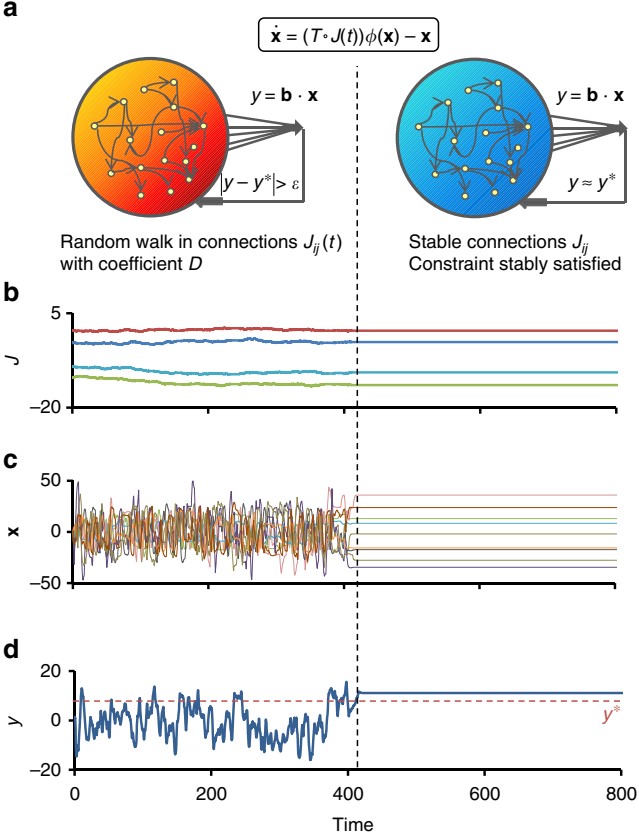

**Figure 1 | Exploratory dynamics and convergence to a constraint-satisfying stable state.** (**a**) Schematic representation of the model: a random $N \times N$ network, composed of an adjacency matrix $T$ and an interaction strength matrix $J$, governs a nonlinear dynamical system (equation in box; $\phi(x) = \tanh(x)$). The resulting spontaneous dynamics are typically irregular for large enough interactions. A macroscopic variable, the phenotype $y$, is subject to an arbitrary constraint $y \approx y^*$ with finite precision $\varepsilon$. When the constraint is not met (left; 'hot' regime), the connections strengths $J_{ij}$ undergo a random walk with magnitude determined by the coefficient $D$ and the mismatch function $\mathcal{M}(y - y^*)$. The random walk stops when the mismatch is stably reduced to zero (right; 'frozen' regime). (**b–d**) Example of exploration and convergence. Shown are representative trajectories of connection strengths (**b**), microscopic variables (**c**) and the phenotype $y$ (**d**) before and after convergence to a stable state satisfying the constraint. The network in this example has scale-free (SF) out-degree distribution ($a = 1$, $\gamma = 2.4$) and Binomial in-degree distribution ($p \simeq \frac{3.5}{N}, N$). $N = 1,000$, $y^* = 10$, $D = 10^{-3}$, $g_0 = 10$. See Methods for more details.

## Results

**Exploratory adaptation model.** To investigate the feasibility of exploratory adaptation, we introduce a model of gene regulatory dynamics incorporating random changes over time in a single network. The model consists of a large number, $N$, of microscopic components $\mathbf{x} = (x_1, x_2 \ldots x_N)$, governed by the following nonlinear equation of motion (Fig. 1a):

$$\dot{\mathbf{x}} = W\phi(\mathbf{x}) - \mathbf{x}, \qquad (1)$$

where $W$ is a random matrix, representing the intracellular network of interactions; $\phi(\mathbf{x})$ an element-wise saturating function restricting the dynamic range of the variables; and the relaxation rates are set to unity. Previous work has used similar equations to address evolutionary aspects of gene regulation[20,21] as well as interactions and relaxation in neuronal networks[22]. Most studies have focused on networks with uniform (full or sparse) connectivity; much less is known about the dynamics for networks with non-uniform topological structures, which may be of relevance to gene regulation.

Here we consider sparse random networks with different types of topological properties. For all cases, the interaction matrix $W$ is composed of an element-wise (Hadamard) product,

$$W = T \circ J, \qquad (2)$$

where $T$ is a random topological backbone (adjacency) matrix with binary (0/1) entries representing potential interactions between network elements; and $J$ is a random matrix specifying the actual interaction strengths. To represent context-dependent regulatory plasticity, we assume that the backbone remains

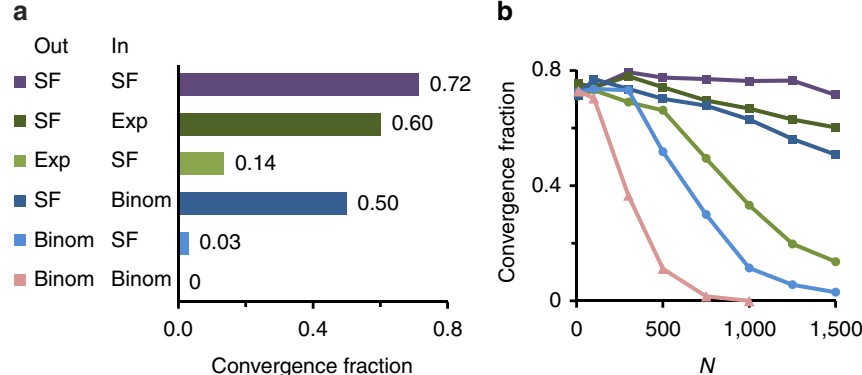

**Figure 2 | Convergence fractions depend on network topology.** (**a**) Seven ensembles of networks of size $N = 1,500$ and different topologies exhibit remarkably different convergence fractions (CFs). Ensembles are characterized by the out- and in- degree distributions of the adjacency matrix $T$: 'SF', scale free distribution; 'Exp', exponential distribution; 'Binom', Binomial distribution. (**b**) CF as a function of network size for the same ensembles of (**a**) with matching colours. $N = 1,500$, $y^* = 0$, $g_0 = 10$, $D = 10^{-3}$. Parameters for degree distributions: SF, ($a = 1$, $\gamma = 2.4$); Binom, $\left(p \simeq \frac{3.5}{N}, N\right)$; Exp, ($\beta = 3.5$).

fixed, whereas the interaction strengths are plastic and amenable to change over time. We will emphasize below network sizes and topological structures that are relevant to gene regulatory networks.

On a macroscopic level we consider a cellular phenotype, $y$, which depends on the microscopic components and can affect the cell's functionality and state of stress. We define this phenotype as a linear combination of microscopic variables

$$y(t) = \mathbf{b} \cdot \mathbf{x}(t) \tag{3}$$

with an arbitrary vector of coefficients $\mathbf{b}$. To model an unforeseen challenge, the system is subjected to an arbitrary contstraint of maintaining the phenotype in a given range $y(t) \approx y^*$. Importantly, any given value of the phenotype can be realized by many alternative microscopic combinations.

Deviation from compliance with the constraint is represented by a global cellular function $\mathcal{M}(y - y^*)$, corresponding to the level of mismatch between the current phenotype and the demand. This mismatch is effectively zero inside a 'comfort zone' of size $\varepsilon$ around $y^*$ and increases sharply beyond it. Biologically, the comfort zone can be interpreted as a range of phenotypes that can be tolerated in a given environment without invoking significant stress. This is represented mathematically by a range of values which satisfy the constraint (in contrast to many optimization problems which require adherence to a specific value).

When the phenotype deviates from the comfort zone, the mismatch drives an exploratory search, realized by small random changes in the interaction strengths, forming a random walk in the elements of the matrix $J$:

$$dJ_t = \sqrt{D \cdot \mathcal{M}(y - y^*)} \cdot d\mathcal{W}_t. \quad J(t=0) = J_0, \tag{4}$$

where $\mathcal{W}_t$ is the standard Wiener process. The amplitude of the random walk is controlled by a scale parameter, $D$, and the mismatch level, $\mathcal{M}$. These random changes can arise from diverse sources of variation affecting the levels of transcription regulators[3,23,24], as well as regulatory interactions (for example, alternative splicing, conformations of transcription factors and their post-translational modifications[17,18]).

The random walk constitutes an exploratory search for network configurations in which the dynamical system in equation (1) satisfies the constraint in a stable manner. Random occurrence of such a configuration decreases the search amplitude, thereby promoting relaxation by reducing the drive for exploration[6,25]. Convergence of this process to a stable state satisfying the constraint is not *a priori* guaranteed. Intuitively, it may be expected that randomly varying a large number of

parameters in a nonlinear high-dimensional system will cause the dynamics to diverge. Surprisingly, we find that the adaptation process can in fact converge; however, as shown below, convergence depends on key properties of the network.

**Adaptation depends on network topology.** An example of adaptive convergence is shown in Fig. 1b–d. At $t = 0$, the system is confronted with a demand and starts an exploratory process in which the connection strengths are slowly modified. Figure 1b displays the time trajectories of four of these connection strengths. During this exploration, the microscopic variables, $\mathbf{x}$, and the phenotype, $y$, exhibit highly irregular behavior, rapidly sampling a large dynamic range (Fig. 1c,d respectively). At $t \sim 400$, the system manages to stably reduce the mismatch to zero and converges to a fixed point (Fig. 1). In some cases the dynamics converges to a small-amplitude limit-cycle (Supplementary Note 3, Convergence to a limit cycle), and remain within the comfort zone $\pm \varepsilon$ around $y^*$. The state of convergence is found to be a stable attractor that is robust against small perturbations of the dynamic variables, $\mathbf{x}$, and the interactions strengths, $W_{ij}$ (Supplementary Note 3, Stability of the adapted state). The differences between the amplitude of temporal changes in Fig. 1b–d reflects the separation of timescales between the slowly accumulating changes in interaction strengths, governed by the small value of $D$ in equation (4), and the intrinsic dynamics of equation (1).

To investigate the dependence of exploratory adaptation on network topology we constructed random matrix ensembles with different topological backbones, manifested by distinct in- and out-going degree distributions[26] (detailed in the Methods section). Each ensemble was evaluated with respect to the probability of convergence, estimated as the fraction of simulations which converged within a given time window. Figure 2a compares ensembles of networks with in- and out-degrees drawn from Binomial (Binom), Exponential (Exp) and Scale-Free (SF) distributions. It shows high fractions of convergence, 0.5 or higher, only for ensembles with SF out-degree distributions. In contrast, the in-degree distribution affects convergence only mildly. For example, the convergence fraction (CF) of networks with SF out-degree and Binomial in-degree distributions (dark blue) is 0.5, whereas it is only 0.03 in the transposed case (light blue). This asymmetry between outgoing and incoming connections indicates that convergence of exploratory adaptation does not rely on spectral properties of the interaction matrix ensemble.

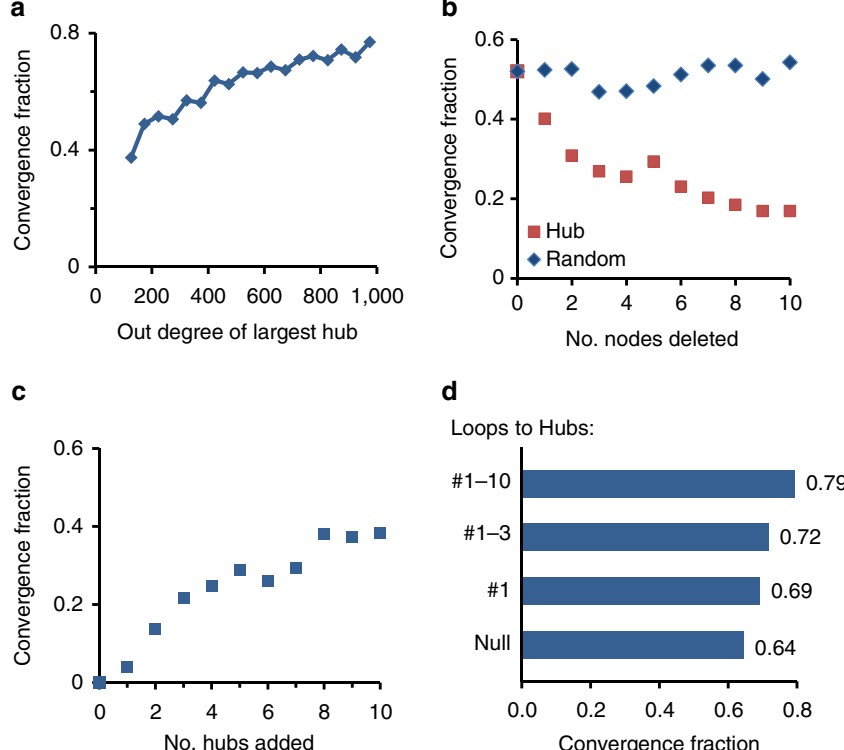

**Figure 3 | Exploratory adaptation depends on the existence of hubs and is enhanced by their auto-regulation.** (**a**) CF versus out-degree of the largest hub in a collection of SF-Binom networks binned according to their largest hub. (**b**) Changes in Convergence Fraction (CF) following deletion of a number of leading hubs (red) or deletion of the same number of random nodes (blue) from networks with SF-Binom topology. (**c**) Effect of adding a small number of outgoing hubs to a Binon-Binom ensemble. The out-degrees of the added hubs were chosen to mimic the SF-out ensemble of Fig. 2. (**d**) Effect of adding autoregulatory loops on a specific number (1, 3 and 10) of the leading outgoing hubs on a background of a SF-Binom ensemble. $N = 1,500$, $y^* = 0$, $g_0 = 10$, $D = 10^{-3}$. Parameters for degree distributions: SF, $(a = 1, \gamma = 2.4)$; Binom, $(p \simeq \frac{3.5}{N}, N)$; Exp, $(\beta = 3.5)$.

Analysis of convergence as a function of network size shows that the effect of topology becomes pronounced for large networks (Fig. 2b). The CF in small to intermediate-sized networks ($N \lesssim 200$) is higher and relatively independent of topology. However, as $N$ increases towards sizes that are relevant to genetic networks, the benefit of having SF out-degree distribution becomes progressively prominent.

**Outgoing hubs enable adaptation in large networks.** Among the topological ensembles tested, an outgoing SF degree distribution was found to be crucial for convergence in large enough networks. Such distributions are characterized by a broad range of heterogeneous connectivities, with a small number of extremely highly connected nodes (hubs). To evaluate the relative contribution of outgoing hubs to convergence within this ensemble, we ranked the backbones of the connectivity matrices drawn from the SF-Binom distributions according to the out-degree of the largest hub. Figure 3a shows that the CF increases with the connectivity of the largest outgoing hub. As a second approach to characterize hub contribution, we deleted a small number of outgoing hubs from these networks[27]; this leads to a significant reduction in CF that is not observed upon removal of randomly chosen nodes (Fig. 3b).

These results indicate that, in networks from the SF-Binom ensemble, outgoing hubs have a major positive influence on the success of exploration. We therefore asked whether the addition of a few hubs to an otherwise poorly converging ensemble is enough to induce significant convergence. Figure 3c indeed shows that addition of as few as 8 hubs to a Binom-Binom ensemble increases the CF from zero to about 0.4.

These findings are in-line with reported properties of gene regulatory networks, particularly the existence of 'master regulatory' transcription factors that control the expression of hundreds of other genes[28–30]. Since many of these master regulators are also autoregulated[31], we evaluated the influence of hub autoregulation on the success of exploratory adaptation in our model. Figure 3d shows that autoregulation of the leading hubs in the SF-Binom ensemble further increases the CFs.

Since autoregulation motifs are commonly observed in gene regulatory networks (not only in hubs)[32], we investigated whether these motifs could also contribute to convergence when over-represented uniformly throughout the network. Figure 4 depicts the results of adding such motifs randomly to 10% of the nodes in networks from the SF-Binom and Binom-SF ensembles. It is seen that positive autoregulation enhances convergence for intermediate sized networks ($N = 1,000$) in both ensembles; this effect is particularly notable for the Binom-SF ensemble, which has small CF without these motifs. This contribution, however, decreases with network size and vanishes in the same type of networks with $N = 3,000$. We conclude that the presence of autoregulatory motifs ranodmly positioned in the network cannot substitute for hub contribution in the limit of very large networks. These results highlight the interplay of several networks properties in exploratory adaptation: network size, topology and autoregulatory motifs. The addition of common network motifs other than autoregulation did not lead to a conclusive effect on convergence (Supplementary Note 3, Dependence of convergence on network motifs).

**Adaptation occurs over a wide range of model parameters.** We investigated how the capacity to adapt is affected by various model parameters. To examine the dependence on the severity of the constraint, we varied the size of the comfort zone $\varepsilon$. Figure 5a reveals a sharp decrease of the CF as $\varepsilon$ is reduced, indicating that a non-vanishing comfort zone is crucial for successful exploratory

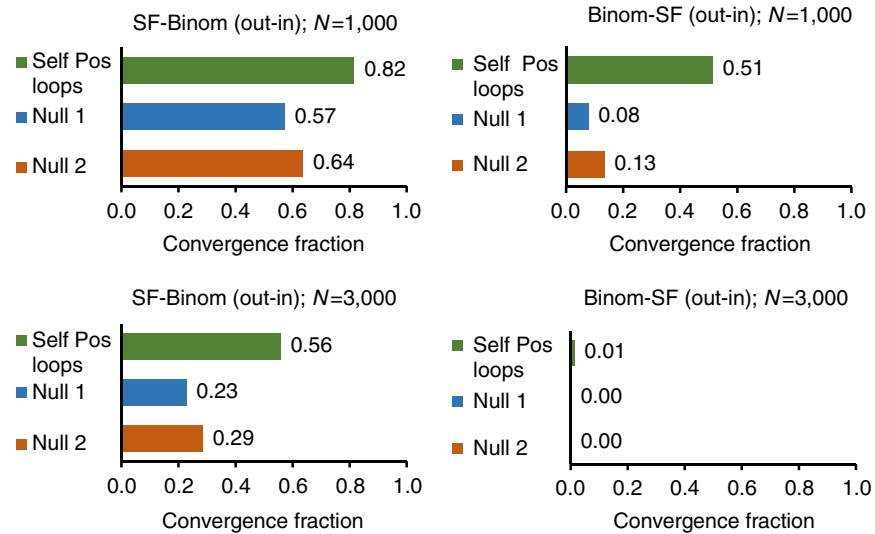

**Figure 4 | Effect of positive autoregulation on convergence fractions.** Positive autoregulatory loops were added randomly to 10% of the nodes in four ensembles, each comprising 500 networks of a given size ($N = 1,000$ or $3,000$) and topology (SF-Binom or vice versa). Convergence in each ensemble is compared to controls without extra loops, with and without matching of the degree distributions to the enriched ensemble (Null 1 and Null2, respectively). Parameters of the SF and Binom distributions (prior to addition of loops) are: SF, ($a = 1$, $\gamma = 2.4$) and Binomial, ($p = \frac{3.5}{N}$ and $N$). Other parameters are $g_0 = 10$, $\alpha = 100$, $\varepsilon = 3$, $c = 0.2$, $D = 10^{-3}$ and $y^* = 0$.

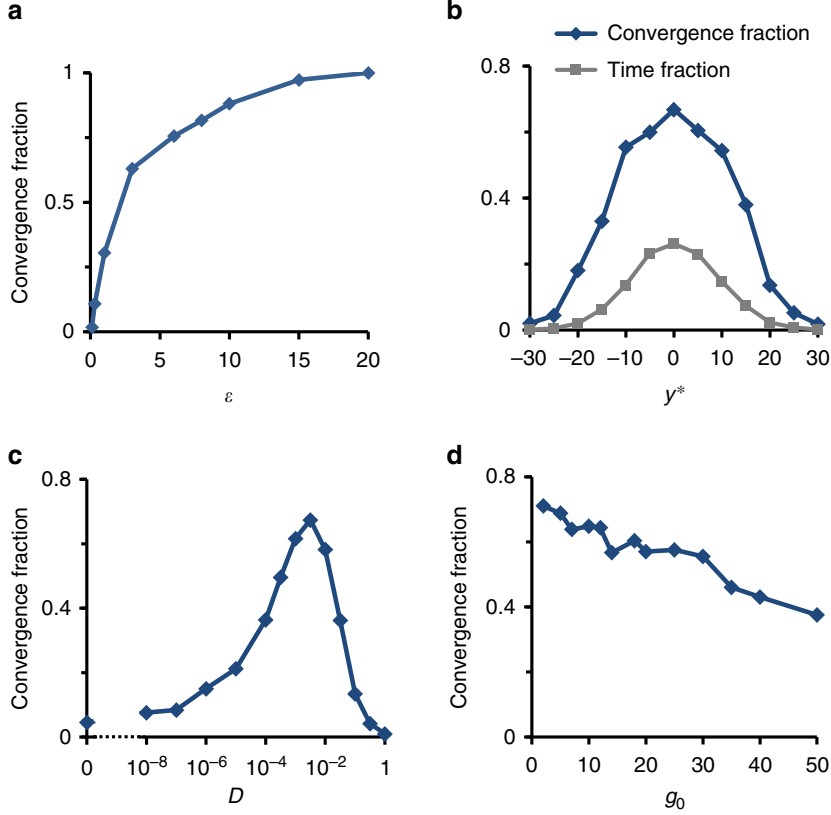

**Figure 5 | Dependence of CF on model parameters.** (**a**) CF versus $\varepsilon$, the width of the comfort zone around $y^*$. (**b**) CF (blue) versus the constraint value $y^*$. For comparison, the grey curve shows the fraction of time in which $y(t)$ spontaneously reaches the constraint-satisfying range. (**c**) CF versus the strength of exploratory random walk in connection strengths, $D$. (**d**) CF versus $g_0$ (proportional to the s.d. of connection strengths; see Methods for details). Network ensemble with SF-out ($a = 1$, $\gamma = 2.4$) and Binom-in ($p \simeq \frac{3.5}{N}$, $N$) degree distributions. Unless otherwise specified, all ensembles have $N = 1,000$, $y^* = 0$, $g_0 = 10$, and $D = 10^{-3}$.

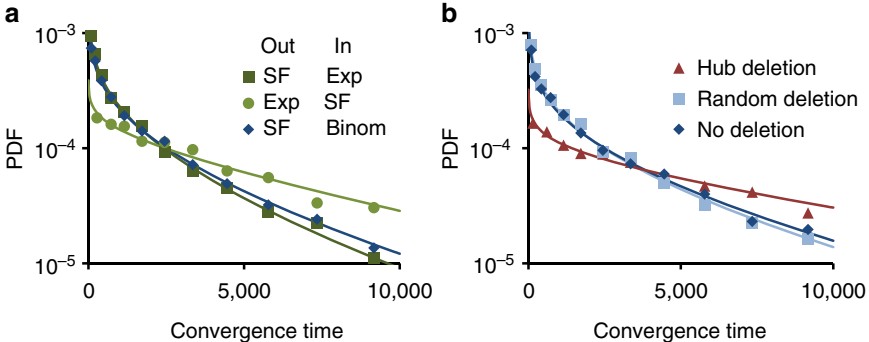

**Figure 6 | Distribution of convergence times for networks which converged in less than $10^4$ timesteps.** Solid lines depict stretched exponential fits. (**a**) Probability density distribution (PDF) of convergence time for three topological ensembles. (**b**) PDFs after deleting the 8 largest hubs (red) or the same number of randomly-chosen nodes (light blue) from the SF-Binom ensemble. All ensembles have $N = 1,000$, $y^* = 0$, $g_0 = 10$, and $D = 10^{-3}$. Degree distribution parameters: SF, $(a = 1, \gamma = 2.4)$; Binom, $\left(p \simeq \frac{3.5}{N}, N\right)$; Exp, $(\beta = 3.5)$.

adaptation. This requirement is biologically plausible, as one expects a range of phenotypes capable of accommodating a given environment rather than a unique optimal phenotype. Another way of increasing the adaptation challenge is by shifting the required phenotypic range away from the origin. Reaching a shifted region is challenging because it is more rarely visited by spontaneous dynamics (Fig. 5b, grey curve). Figure 5b indeed shows that the CF decreases as $y^*$ moves away from zero (blue curve). Importantly however, it remains much larger than the probability of encountering the required phenotype spontaneously. For example, a non-negligible convergence (CF $\sim 0.2$) is observed even for an interval around $|y^*| \sim 20$ which is spontaneously encountered with probability of 0.02.

To evaluate the sensitivity of adaptation to exploration speed, we varied the effective diffusion coefficient in the space of connection strengths, $D$. Figure 5c shows that a non-zero convergence fraction is achieved for a wide range of this parameter and remains between 0.2 and 0.7 over more than 5 orders of magnitude. As the value of $D$ increases beyond a certain level where the separation of timescales ceases to hold, the convergence fraction decreases rapidly.

For a given adjacency matrix $T$, interactions within the network are determined by the connections strengths, $J_{ij}$. These are initially drawn from a Gaussian distribution with a zero mean and a given s.d. The s.d. normalized to network size, $g_0$ (also called network gain; for details see Methods) determines the contribution of the first versus second term in equation (1). In large homogeneous networks, this parameter has a strong effect on the dynamics of equation (1) (ref. 33). In contrast, we find that the capacity to adapt by exploration in our model is relatively weakly dependent on $g_0$ (Fig. 5d).

**Broad non-exponential distributions of adaptation times.** The analysis presented so far was based on convergence fractions within a fixed time interval. To characterize the temporal aspects of exploratory adaptation, we evaluated the distribution of convergence times in repeated simulations. Figure 6 reveals a broad distribution (CV $\approx 1.1$), well fitted by a stretched exponential (see Supplementary Note 3, Stretched exponential fit to the distribution of convergence times). Such distributions are common in complex systems[34] and were suggested to reflect a hierarchy of timescales[35]. While the general shape of the distributions were similar in all topological ensembles tested, networks with SF out-degree distributions typically converged faster than their transposed counterparts (Fig. 6a). Moreover, deletion of a small number of leading outgoing hubs causes a

significant shift towards longer convergence times (Fig. 6b). Thus, networks with larger heterogeneity in out-degrees are both more likely to converge within a given time window (Figs 2 and 3), and typically converge faster (Fig. 6).

**Adaptation success correlates with abundance of attractors.** In the typical example shown in Fig. 1, exploratory dynamics culminates in reduction of drive for exploration and convergence to a stable attractor of equation (1). The significant differences between adaptive performance of network ensembles (Fig. 2a,b) may reflect the abundance of networks supporting relaxation to attractors in the different ensembles. Previous work has shown that for networks with uniform degree distributions and sufficiently strong interactions, the number of attractors of equation (1) decreases with network size and vanishes in the limit of infinite size (leading to chaotic motion only)[33]. A related result was recently found for Boolean networks[36]. It is not known, however, how the number of attractors scales with system size for networks of arbitrary topological structure.

To address this question, we simulated many independent networks in each ensemble and estimated the fraction which relaxed to fixed points without exploration or feedback (equation (1) alone). For any given network, the probability of relaxation to a fixed point was largely insensitive to the initial conditions in **x**-space (not shown). With that in mind we computed, for each topological ensemble, the fraction of networks supporting relaxation within a given time window, starting with random initial conditions. This measure is analogous to the CF used in Fig. 2, but without any constraint, feedback or random walk in connection strengths. To highlight the dependence on network size we extended the simulations up to $N = 10,000$. Figure 7a reveals topology-dependent differences that are qualitatively in line with the ability for exploratory adaption shown above (Fig. 2b). This suggests that a substantial contribution to successful adaptation is indeed provided by a high abundance of networks exhibiting fixed points in their dynamics.

For each network ensemble that supports fixed points, we further analysed the distribution of relaxation times into these fixed points. Figure 7b demonstrates the effect of topology by comparing the SF-Exp ensemble to the transposed Exp-SF. It shows that networks with SF-out degree distribution typically support faster relaxation to their respective fixed points. This may allow the adaptation to converge before exploration has had a chance to significantly modify network connections. Further work is required to test this hypothesis and to broaden the

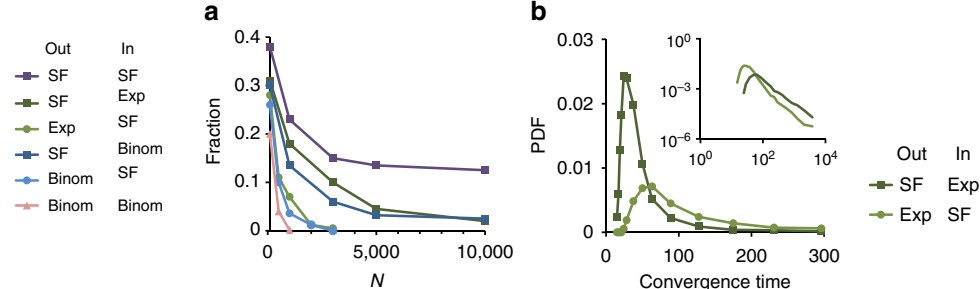

**Figure 7 | Fixed points in the absence of exploration for different topological ensembles.** (**a**) Fraction of networks within an ensemble which relaxed to a fixed point under the nonlinear dynamics of equation (1), with fixed connections, no constraint and no feedback. Topological ensembles which exhibited higher success in exploratory adaptation in Fig. 2b, relaxed to fixed points in a larger fraction of simulations. (**b**) Distribution of relaxation times to fixed points for two of the ensembles. Note the shorter typical timescale for the SF-Exp ensemble (the more successfully adapting ensemble). $N = 1,000$, $g_0 = 10$. Dergree distribution parameters: SF, $(a = 1, \gamma = 2.4)$; Binom, $\left(p \simeq \frac{3.5}{N}, N\right)$; Exp, $(\beta = 3.5)$.

theoretical understanding of these dynamics in random ensembles with heterogeneous topologies.

## Discussion

Overall, we have introduced a model of exploratory adaptation driven by mismatch between an internal global variable and an external constraint. Adaptation is achieved by a purely exploratory process which relies on the plasticity of regulatory interactions[17,18]. Our model was described in terms of gene regulation but could equally well represent adaptation in other cellular interactions, such as the protein-protein interaction network. We have found that convergence of exploratory adaptation depends crucially on structural properties of the network. It requires the existence of outgoing hubs and is enhanced by auto-regulation of these hubs. These results offer an important, but hitherto unrealized, rationale for the overwhelming abundance of autoregulation motifs on master regulatory transcription factors[31]. These master regulators act as network hubs by virtue of the large numbers of their downstream gene targets. Our findings show that autoregulation of such hubs improves their ability to drive the network into a stable state which satisfies a phenotypic demand.

The contribution of outgoing hubs to the success of adaptation may reflect their ability to coordinate changes in a large set of affected nodes. In a network with a narrow distribution of out-degrees (without hubs), each node has the same relatively small influence as any other node. In the absence of a hierarchy in the extent of influence, irregular dynamic variation in the microscopic variables is unlikely to accumulate into a macroscopic coherent change in phenotype. On the other hand, the existence of a few hubs with a much broader influence can promote correlations between many downstream nodes, leading to an ability to drive a coherent change in a given direction. These effects may be related to other aspects of stability in network dynamics that vary with topology[37–39].

Beyond the structural aspects promoting exploratory adaptation, the process of convergence appears to be complex and is characterized by an extremely broad distribution of times. Successful convergence likely depends on a delicate interplay between the space of possible network configurations, their connectivity properties and the typical timescales of their intrinsic dynamics.

While our model draws from neural network models[40–42], it is substantially different in relying on purely stochastic exploration. In the language of learning theory, the 'task' is modest: convergence to a stable attractor which satisfies a low-dimensional approximate constraint. Without exploration, this task could be fulfilled by chance with a very small probability. This probability increases dramatically by exploratory dynamics within a class of networks of a given structure. The ability to achieve high success rates without a need for complex computation or fine-tuning makes this type of adaptation particularly plausible for biological implementation. The relevance of similar processes in neural networks remains to be investigated.

Random network models were previously used to address evolutionary dynamics of gene regulation over many generations. These studies considered a population of networks undergoing random mutations and selection according to an assigned fitness[21,43]. In contrast, the model presented in the current study considers random variations over time within a single network, as an abstraction of a particular aspect of single cell adaptation within its lifetime. While these two approaches differ in timescales, level of organization and biological phenomena, it seems that they cannot be completely decoupled and that biological networks have basic properties that reflect on both contexts[44]. For example, in the context of selection in a population of networks, marked differences in evolutionary dynamics were found between homogeneous and SF networks[45]. In fact, the reproducible and exploratory responses in single cells, and the evolutionary processes at the population level, correspond to complementary aspects of gene-environment interactions at different scales[3,46]. A major future goal would be to integrate these aspects into a general picture of adaptive responses to diverse types of challenges over a broad range of timescales.

## Methods

**Constructing network backbone T for topological ensembles.** Interactions between the intracellular dynamical variables are governed by the network matrix $W$, defined as the element-wise (Hadamrd) product of the binary backbone, the adjacency matrix $T$, and a Gaussian random matrix $J$ of connection strengths (equation (2)). We construct an ensemble of a given topology by sampling the connectivities of the backbone from a particular choice of in-degree and out-degree distributions, $P_{in}(K^{in})$ and $P_{out}(K^{out})$, and by sampling the random strengths of $J$ independently from a Gaussian distribution. In practice, $T$ is constructed first by randomly sampling a list of $N$ out-going degrees $\{d_i^{out}\}_{i=1}^N$ from the distribution $P_{out}(K^{out})$ with $d_i^{out} \leq N - 1$; and then sampling a list of $N$ in-coming degrees $\{d_i^{in}\}$ from the distribution $P_{in}(K^{in})$ (again $d_i^{in} \leq N - 1$), conditioned on the graphicality of the in- and out- degree sequences[47]. The network is then constructed from these sequences using the algorithm described in[48].

Scale-free (SF) sequences are obtained by a discretization to the nearest integer of the continuous Pareto distribution $P(K) = \frac{(\gamma - 1)a^{(\gamma - 1)}}{K^\gamma}$. Sampling SF degree sequences using the discrete Zeta distribution gives qualitatively similar results. Binomial sequences are drawn from a Binomial distribution $P(K) = \mathcal{B}(N, p)$, with $p = \frac{\langle K \rangle}{N}$. Exponential sequences are obtained by a discretization to the nearest integer of the continuous exponential distribution $P(k) = \frac{1}{\beta} e^{-\frac{K}{\beta}}$ with $\beta = \langle K \rangle$. A Binomial degree sequence is implemented using MATLAB Binomial random number generator. Exponential and Scale-free sequences are

implemented by a discretization of the continuous MATLAB Exponential and Generalized Pareto random number generators with parameters $k = 1/(\gamma - 1)$, $\sigma = a/(\gamma - 1)$ and $\theta = a$.

**Comparison between different ensembles.** To compare adaptation performance between different ensembles, interaction matrices need to be properly normalized. In the study of uniform random matrices, the elements are usually normalized such that their variance is $\frac{g_0^2}{N}$, providing a well-defined thermodynamic limit $N \to \infty$ in which the matrix eigenvalues of are uniformly distributed within a disc of size $g_0$ in the complex plane[49,50].

In our model, the initial interaction matrix $J_0 \triangleq J(t=0)$ is defined as a random Gaussian matrix with mean 0 and variance $\frac{g_0^2}{\langle K \rangle}$, $\langle K \rangle$ being the average connectivity. Neglecting correlations in the adjacency matrix $T$, the variance of its elements is $Var(T_{ij}) = \frac{\langle K \rangle}{N}(1 - \frac{\langle K \rangle}{N}) \approx \frac{\langle K \rangle}{N}$, which implies $Var(W_{ij}) \approx \frac{g_0^2}{N}$. In principle both finite-size effects and correlations in $W_{ij}$ result in deviations from a uniform distribution of eigenvalues in the circle. However empirically we find that for matrices of relevant size, the spectral radius of $W$ is still $\sim g_0$, establishing a basis for comparison between the different ensembles based on spectral radius. We note however that the eigenvalue distribution is far from being uniform (see Supplementary Note 1, Empirical spectrum of interaction matrices $W$).

Another model component that needs to be normalized for proper comparison is the macroscopic phenotype $y(\mathbf{x}) = \mathbf{b} \cdot \mathbf{x}$. The arbitrary weight vector $\mathbf{b}$ is characterized by a degree of sparseness $c$, the fraction of non-zero components, $\frac{1}{N} < c < 1$; and by the typical magnitude of those components. In order to compare between networks of different size and weight vectors of different sparseness, the variance of the non-zero components is scaled by their number, $cN$ and by the matrix gain $g_0^2$. The non-zero components of $\mathbf{b}$ are thus distributed as $b_i \sim \mathcal{N}(0, \frac{1}{g_0^2 \cdot cN} \cdot \alpha)$, where $\alpha$ is a single parameter determining the scale of phenotype fluctuations in different network sizes and gains (See Supplementary Note 1, Distributions of phenotype $y$).

**Computing convergence fractions.** Convergence fractions were computed over 2,000 time steps in samples of 500 networks drawn from specified in- and out-degree distributions, averaging over $T$, $J_0$ and $\mathbf{x}_0$. For fully or sparsely connected homogeneous random networks of size $N = 1,500$, the CF is close to zero (not shown). Alternative ensemble definitions (for example, keeping $T$ fixed) do not change the main results (see Supplementary Note 1, Convergence of different network ensembles).

**Saturating function $\phi(\mathbf{x})$.** The saturating function is defined as an element-wise function $\phi(x_j) = \tanh(x_j)$ operating separately on each of the components of $\mathbf{x}$. Model results are insensitive to the exact shape of this function (Supplementary Note 2, Robustness of model to saturating function $\phi$) and to placing the saturation inside or outside of the interactions (Supplementary Note 2, Robustness of model to position of saturating function $\phi$).

**Mismatch function $\mathcal{M}(y - y^*)$.** The mismatch function is defined here as $\mathcal{M}(y - y^*) = \frac{\mathcal{M}_0}{2}\left[1 + \tanh\left(\frac{|y - y^*| - \varepsilon}{\mu}\right)\right]$, a symmetric sigmoid around $y^*$, where $\varepsilon = 3$ controls the size of the low-mismatch 'comfort-zone' around $y^*$, $\mu = 0.01$ the steepness of the sigmoid and $\mathcal{M}_0 = 2$ its maximal value. Main model results are insensitive to the exact shape of this function as long as it has a flat region with zero or very low mismatch around $y^*$. (see Supplementary Note 2, Robustness of model to mismatch function $\mathcal{M}$).

**Data availability.** The data that support the findings of this study are available from the corresponding author upon reasonable request.

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

## Acknowledgements

We thank O. Barak, E. Braun, R. Meir and M. Stern for valuable discussions and S. Marom, A. Rivkind, L. Geyrhofer and H. Keren for critical reading of the manuscript.

## Author contributions

Y.S. and N.B. conceived the general approach for modelling adaption by exploratory dynamics. H.I.S. and N.B. constructed the model. H.I.S. performed all the simulations and computations. All authors evaluated model findings and designed simulations to identify requirements and properties of exploratory adaptation. All authors wrote the manuscript.

## Additional information

**Competing interests:** The authors declare no competing financial interests.

**Publisher's note**: 

