## [Peer review file · Nature Communications]

Reviewers' comments:

Reviewer #1 (Remarks to the Author):

I have reviewed the manuscript entitled “Exploratory Adaptation in Large Random Networks” written by Schreier et. al. The authors studied a phenomenological model (known as “circuit” equations in the neuron network literature) and explored the impact of network topology on the convergence rate of exploratory adaptation. Their key result is that networks with a scale-free out-degree distribution are much more likely to converge. This result is consistent with some experimental findings.

I find that the research topic is very interesting. And to some extent the presented results are compelling. Yet, I do have several comments on the current form of the manuscript. I am quite open to looking at a revised version, if the authors can address in a satisfactory fashion the issues discussed below:

Model:

1. It is unclear if the presented results rely heavily on the details of the phenomenological model. For example, what happens if the first term, i.e. $W \phi(x)$, in the r.h.s of Eq.(1) is replaced by $\phi(Wx)$?

2. What's the functional form of the saturating function $\phi(x)$? Will the results depend on $\phi(x)$?

3. The mismatch function $M(y-y^*)$ is defined to be zero inside a “comfort zone” around the desired phenotype y^* and increases sharply beyond this zone. The authors mentioned in the SI that the existence of the comfort zone (rather than a well-defined minimum at a point) seems to be essential for convergence by exploratory adaptation. What's the biological interpretation of this “comfort zone”? Do we have any experimental evidence for its existence?

4. In some other modeling frameworks of gene regulatory networks, e.g. Boolean network, cell types and cellular functional states are naturally associated with the intrinsic attractors of the Boolean networks (without any exploratory adaptations). Does the gene regulatory dynamical model studied in this paper also support various intrinsic attractors in the absence of exploratory adaptation?

Results:

5. The authors provide neither analytical nor intuitive/qualitative explanations why the adaptation processes can in fact converge and why networks with a scale-free topological backbone are much more likely to converge.

6. How robust are the stable connections against small perturbations on the interaction strengths J and the topology backbone T ?

7. It is well known that gene regulatory networks contain certain motifs, e.g. feed-forward loop and bi-fan. Will the presence of those motifs enhance the exploratory adaptation? If yes, this will make the phenomenological model more appealing. Otherwise, we have to

rethink the meaning of this model.

Reviewer #2 (Remarks to the Author):

A. In this contribution, the authors consider a network model that couples regulatory genes and metabolic genes, to study whether and how such networks could explore a fitness landscape in response to an altered environment—but non-genetically, rather, during its lifetime. The methods they use are inspired to some extent by cognitive learning algorithms, which use similar constructions.

. B To a large extent, this is a study that explores the feasibility of such a learning algorithm, using coupled networks. Within computational biology, the networks the authors use are fairly standard, but their coupling is, to my knowledge novel (at least outside this group of authors). In essence, the idea is that the fixed point of the network (assuming it exists) must move to a new location in response to a changed environment. The authors achieve this by implementing a random walk on the network edge weights, and using a fitness function that has its maximum near the new fixed point. They find that networks can converge to the new fixed point, but that the rate of convergence depends significantly on the backbone architecture. In particular, scale-free networks are “best” in this respect.

C. Remarkably, the only evidence that such an algorithm may be at work in biological organisms that the authors present is precisely this finding, namely that regulatory networks with a scale-free out-degree distribution do this learning best. But even this evidence is, in my view, fairly flimsy. The field of systems biology has an odd fascination with scale-free distributions, and if they can fit a straight line to a log-log plot even if it only covers a single decade, then they declare success. But the data (for example that presented in [22]) can really be fit in multiple ways. To make the case for a scale-free distribution really requires at least two decades, better 2.5. So for me, the finding that networks with SF out-degree distribution and exponential in-degree distribution do best is at best a hint that the model the authors present is not necessarily completely wrong.

D. NA

E. So this is my main problem with this paper: it is completely speculative, with essentially no hint given as to why we should believe that this mechanism is active in biology. I am not saying that this mechanism can't exist in biology. It absolutely could. Something like this may be going on in cognitive networks but (I also work in that field) this is by no means established. Indeed, the ANN abstraction in computational neuroscience is to a large extent laughable. The idea that behavioral responses are encoded in fixed points of a high-dimensional manifold is easily repudiated by finding that small changes in the input pattern in brains can create dramatic changes in the response, something that is not possible in the

ANN picture. The continuous response is, in fact, related to the connectivity pattern of the ANNs, something that the present work is changing by using sparse networks, instead. This is indeed the correct approach.

F. I'm afraid that in the absence of more biological evidence, a paper like this belongs in a far more specialized journal. Just as an example, the authors describe a random walk procedure that modifies the weights of the network, but no example of a biological mechanism that could create this kind of noise is offered. It is true that transcription, e.g., can be noisy. But there is no evidence that transcription rates can be shifted to new levels when exceeding some threshold, say. This could happen if gene expression was bistable (which can happen sometimes). But it is this kind of molecular evidence that the kind of dynamics that are being proposed in this paper that needs to be marshaled in order to be published in this venue. In the absence of those, a more specialized journal is more appropriate.

Signed: C. Adami

Reviewer #3 (Remarks to the Author):

The authors propose an interesting approach to evolutionary adaptation, where a cell (or any other complex system) adapts to perform a new function (phenotype) through gradual changes in its interaction strengths. They use this framework to investigate the network characteristics that increase or decrease the system's adaptability. Along the way they find, rather strikingly in my opinion, that the optimal network structures are precisely those exhibited by real gene regulatory networks, namely scale-free out-degree distribution vs. bounded in-degree distribution. I find the proposed model, and especially the results pertaining to gene-regulatory networks to be very interesting - indeed, exposing another hidden role of the commonly observed scale-free property. I also think the paper is clearly written and well-motivated, hence I would be very happy to see it published in Nature Communications. I include my comments below, to help further improve the paper:

1. The authors exemplify their method on a rather general dynamic equation (1), with the main restriction being that the interaction function $\phi(x)$ asymptotically saturates for large x . While this is a rather common assumption in the context of gene-regulation, as the authors clearly note and appropriately cite, one still wishes to better understand the implications of such restriction. What would happen in case we implement non-saturating interaction dynamics? Would that harm the reported findings? It seems to me that saturating vs. divergent dynamics represents an essential distinction, with likely significant implications on the system's behavior, which may potentially enrich the picture currently portrayed.
2. On that note, seems that the main motivation in the presented analysis relates to gene regulation, which follows the combined scale-free/bounded degree distribution, and likely the saturating $\phi(x)$. Cellular function, however is also driven by protein interactions - an undirected scale-free biochemical interaction network with potentially non-saturating interaction terms. I do not feel that the paper needs a full-scale analysis of such interactions, it is rather rich and comprehensive in its current form, but, the authors may consider addressing this

issue briefly, to better put their results in context.

3. In Eq. (3) it seems that the phenotype $y(t)$ is time dependent, namely that the function to which the cell should evolve is not just a point in "phenotype-space", but a complete temporal function. Upon first reading I could not fathom how a system could ever successfully adapt to such a requirement, seeking a target network that not only satisfies a desired fixed point, but also features a specific temporal pattern. Reading on, I realized, that this was not the authors' intention, and indeed, the adaptation leads the system to a desired fixed point. This point should be made clear in the paper, and especially in the presentation of Eq. (3).

4. The exploration follows a form of diffusion, depending on the square-root of the discrepancy from the desired phenotype. How important is this assumption to the convergence? The other arbitrary dependence is on the value of D , which the authors analyse comprehensively later on.

5. The one major issue that begs for additional insight is the way in which the degree distribution affects the adaptation process. The finding that a combined scale-free out-degree and bounded in-degree increases the probability of successful adaptation is central to this paper, and truly important. Hence rather than an intriguing empirical fact, the reader expects to gain more insight into the microscopic/mechanistic exchanges that enable this. This can probably be done through extensive numerical analysis, tracking the specific exploration steps that led to successful adaptation, and observing their relationship to the hubs. Even in the absence of an analytical explanation, I am certain, that such an analysis will also, along the way, shed light on the presented adaptation model and expose new facets and directions for researching this problem.

6. Minor corrections: Two citations misplaced (3,19) below Eq. (4); Page 3: "...substantially more abundant than" - should be that.

To summarize, I believe this is a very good paper, that can be further improved through few "adaptations", if the authors expand on the points above.

Response to Reviewer's comments

Reviewer #1:

Model:

1. It is unclear if the presented results rely heavily on the details of the phenomenological model. For example, what happens if the first term, i.e. $W \phi(x)$, in the r.h.s of Eq.(1) is replaced by $\phi(Wx)$

2. What's the functional form of the saturating function $\phi(x)$? Will the results depend on $\phi(x)$?

We addressed these two related comments with new simulations evaluating the effects of different saturating functions and their placement in equation 1. As shown in new Supplementary Figures S4 and S5 (new Supplementary Sections 2.1 and 2.2), these modifications do not alter the conclusions of the paper, thus providing support to the model's robustness to details.

3. The mismatch function $M(y-y^)$ is defined to be zero inside a “comfort zone” around the desired phenotype y^* and increases sharply beyond this zone. The authors mentioned in the SI that the existence of the comfort zone (rather than a well-defined minimum at a point) seems to be essential for convergence by exploratory adaptation. What’s the biological interpretation of this “comfort zone”? Do we have any experimental evidence for its existence?*

Biologically the comfort zone can be interpreted as representing the degeneracy of organism’s function or fitness with respect to its phenotype. Although the environment imposes a "demand" on a certain phenotype, this demand can be satisfied by a non-zero range of phenotypic values before invoking a significant stress. Support for this view comes from the broad phenotypic variability in populations of indistinguishably-fit cells and multicellular organisms. It is, in fact, widely accepted that no living organisms can avoid a certain level of variability (regardless of how fit it is). This feature makes the model particularly suitable for biological systems and distinguishes it from standard mathematical optimization problems. We have revised the manuscript to clarify this point.

We note also that the mismatch need not be strictly zero inside the comfort zone but small enough. In particular in one implementation we used a sharp sigmoid as a mismatch function (Supplementary Section 1.4).

4. In some other modeling frameworks of gene regulatory networks, e.g. Boolean network, cell types and cellular functional states are naturally associated with the intrinsic attractors of the Boolean networks (without any exploratory adaptations). Does the gene regulatory dynamical model studied in this paper also support various intrinsic attractors in the absence of exploratory adaptation?

This is a central issue in Boolean networks which is currently at the focus of much interest. Our model contributes to this field by showing that attractors do exist in certain topological structures of strongly coupled networks in high dimensions.

It was recently demonstrated [1] that the number of fixed points in Boolean networks decreases dramatically as network size increases, reaching a tiny fraction ($\sim 10^{-3}$) for sizes of few thousand, relevant to small regulatory networks (e.g. of bacteria) and an even smaller fractions for larger networks (e.g. mammalian cell). Beyond Boolean networks, it is known that application of our equation (1) to large **homogeneous** networks with strong enough coupling leads to chaotic motion without attractors [2]; thus the number of attractors decreases to zero in the limit of large networks.

Our current findings (Fig. 5) indicate that this does not necessarily hold for arbitrary topological network structure. Specifically, we show that in the absence of exploration, the number of attractors in an ensemble is strongly dependent on its topology and decreases extremely slowly or reaches a plateau for heterogeneous outgoing degree distributions.

For our context, these results show that the number of attractors correlates positively with the ability of the ensemble to support exploratory adaptation.

Following the reviewer's remark, we revised the text to emphasize and discuss this finding and its relation to other modeling frameworks (e.g. Boolean networks).

Results:

5. The authors provide neither analytical nor intuitive/qualitative explanations why the adaptation processes can in fact converge and why networks with a scale-free topological backbone are much more likely to converge.

Following the reviewer's suggestion, we include a plausible intuitive/qualitative explanation in the revised manuscript. It is based on the differential ability of hubs to coordinate changes in the state of nodes with lower out-degrees. Without hubs (e.g. in a network with a narrow distribution of out-degrees), each node has the same small influence as any other node. Under irregular dynamics, the states of these nodes are expected to undergo random changes, and in a large enough network they are highly unlikely to move coherently towards a stable alteration in a (macroscopic) phenotype. On the other hand, the existence of a small number of nodes with a much broader influence (hubs) promotes correlations between many downstream nodes, leading to a substantial increase in the ability to encounter a coherent change in a given direction. Convergence to a new phenotype which satisfies the demand can then be achieved if the state of one or few hubs changes in an extent and direction that are compatible with this demand.

Preliminary support in this argument is obtained by computing the Pearson correlation between pairs of variable during the irregular dynamics that precedes convergence. Below are shown the distributions of values obtained from the two opposite ensembles, Binom-SF and SF-Binom (out-in). It can be seen that the latter shows a broader range of correlation coefficients, and particularly a subset of variables show extremely high (-1 or 1) correlation. Further work is required to establish the role of hubs in this process and to characterize the nonlinear dynamics quantitatively. This work is currently under way.

6. How robust are the stable connections against small perturbations on the interaction strengths J and the topology backbone T ?

We addressed this important question with new data displayed in 3 supplementary figures (Figs. S11-S13) and described in a dedicated Supplementary section 2.8. Altogether, these data demonstrate the existence of a large basin of attraction for convergence in both J and T space, indicating that the connections are indeed stable for small perturbations in connection strength and topology backbone. We also note this important insight in the revised manuscript (p. 5, l. 129).

7. It is well known that gene regulatory networks contain certain motifs, e.g. feed-forward loop and bi-fan. Will the presence of those motifs enhance the exploratory adaptation? If yes, this will make the phenomenological model more appealing. Otherwise, we have to rethink the meaning of this model.

This is again a very important issue which we have addressed with extensive analysis. We tested the effect of adding common types of network motifs to a random network ensemble (this was done with special care to compare the results to an appropriately-defined null model). The most significant results were obtained for the auto-regulatory feedback loop motifs. We found that even a single auto-regulatory loops acting on the largest hub can lead to a significance enhancement of convergence! This enhancement is further increased by adding such loops to additional hubs. These results are now presented in a new figure panel in the revised version, Fig. 2F. Addition of a sufficient number of loops to random elements can also improve the convergence of small enough networks with narrow degree distributions, but this improvement vanishes in larger networks (new Supplementary Fig. S15).

These new findings are not only consistent with recent results on the contribution of auto-regulation to stability in Boolean networks [3]; they offer an important, but hitherto unrealized, rationale for the overwhelming abundance of positive autoregulation of master regulatory transcription factors (summarized in [3]). Due to the large number of downstream targets of these regulators, they provide an example of hubs in gene regulatory networks. Our results show that autoregulation of such hubs leads to a dramatic improvement in their ability to drive the network into a stable state that is compatible with a phenotypic demand. This is now highlighted in the revised paper.

In contrast to this remarkable impact, we found that addition of Feed-Forward Loops (FFL) has only a mild influence on the convergence fraction and that Bifans have no detectable effect. This is possibly because this ensemble has a scale-free outgoing connection distribution, which already favors both of these motifs [4, 5].

A detailed account of the motif analysis is given in a new Supplementary Section, 2.9.

We are grateful to this reviewer for his/her constructive comments and suggestions which have greatly improved our paper.

Reviewer #2:

C. Remarkably, the only evidence that such an algorithm may be at work in biological organisms that the authors present is precisely this finding, namely that regulatory networks with a scale-free out-degree distribution do this learning best. But even this evidence is, in my view, fairly flimsy. The field of systems biology has an odd fascination with scale-free distributions, and if they can fit a straight line to a log-log plot even if it only covers a single decade, then they declare success. But the data (for example that presented in [22]) can really be fit in multiple ways. To make the case for a scale-free distribution really requires at least two decades, better 2.5. So for me, the finding that networks with SF out-degree distribution and exponential in-degree distribution do best is at best a hint that the model the authors present is not necessarily completely wrong.

We agree with the reviewer's point that power-laws are sometimes inferred without sufficient empirical data. Our model, however, does not assume an underlying power-law distribution of real gene-regulatory networks. We only use the Scale-Free ensemble as a mathematical tool to compare different network statistical properties in a controlled way. In our simulations, as in real networks, the sample of ~1500 connectivity values within a network clearly cannot follow precisely a power-law. The relation of our work to the claimed power-law distributions in experiments may have been overstated and we modified this in the revised text.

Notwithstanding this point, the reviewer's comment motivated us to address the question whether our main findings are sensitive to having exact power-law connectivity. For that, we investigated in more detail the influence of one or few hubs alone. Since hubs are unquestionably abundant in gene-regulatory networks, a significant contribution of a few hubs to exploratory adaptation can provide evidence for biological relevance. To distinguish the effects of hubs from the strict requirement of scale-free topology, we analyzed the effect of adding a small number of hubs to a network with an otherwise narrow distributions of out-degrees. As shown in the new Fig. 2E, the existence of a small number of hubs is sufficient to confer substantial convergence on an otherwise non-converging ensemble. This shows that the feasibility of adaptation by random exploration is not dependent on the exact distribution shape. Rather it is robustly promoted by the existence of outgoing hubs, whose biological existence and importance is beyond any doubt.

E. So this is my main problem with this paper: it is completely speculative, with essentially no hint given as to why we should believe that this mechanism is active in biology. I am not saying that this mechanism can't exist in biology. It absolutely could. Something like this may be going on in cognitive networks but (I also work in that field) this is by no means established. Indeed, the ANN abstraction in computational neuroscience is to a large extent laughable. The idea that behavioral responses are encoded in fixed points of a high-dimensional manifold is easily repudiated by finding that small changes in the input pattern in brains can create dramatic changes in the response, something that is not possible in the ANN picture. The continuous response is, in fact, related to the connectivity pattern of the ANNs, something that the present work is changing by using sparse networks, instead. This is indeed the correct approach.

The revised manuscript includes new findings, demonstrating that the (well-known) existence of a few hubs is sufficient to facilitate exploratory adaptation (new Fig. 2E) and that this adaptation is further enhanced by autoregulation, an abundant feature of gene regulatory hubs (new Fig. 2F; new Supplementary section 2.9). These results are completely free of speculation, thus eliminating any potential doubt about the biological relevance of our model.

Beyond this demonstration of biological relevance, we would like to emphasize that the purpose of this work is to suggest the first conceptual explanation for a class of currently unexplained biological phenomena. This is the main novelty of our work and accordingly this is how it should be judged. Establishing this model with sufficient phenomenological evidence requires experimental effort, which is clearly beyond the aim or scope of this theoretical work.

We agree with the reviewer that this work may have implication to learning theory beyond the standard ANN (as indeed is reflected by the interest we receive from colleagues in the Neuroscience community). Rather than encoding responses in attractors, the system finds a different attractor state each new encounter; a nontrivial task in high dimensions. This could be compatible with the described behavior.

F. I'm afraid that in the absence of more biological evidence, a paper like this belongs in a far more specialized journal. Just as an example, the authors describe a random walk procedure that modifies the weights of the network, but no example of a biological mechanism that could create this kind of noise is offered. It is true that transcription, e.g., can be noisy. But there is no evidence that transcription rates can be shifted to new levels when exceeding some threshold, say. This could happen if gene expression was bistable (which can happen sometimes). But it is this kind of molecular evidence that the kind of dynamics that are being proposed in this paper that needs to be marshaled in order to be published in this venue. In the absence of those, a more specialized journal is more appropriate.

There is in fact ample evidence for irregular (random) activity in gene regulatory networks that can support the type of random walk that we model. Transcriptional noise is only part of the picture: it can providing changes in regulation through modifying the expression of transcription factors. Additionally, context-dependence regulation, implemented by intrinsically disordered transcription factors; the practically endless combinatorics of alternative splicing; and multiple post-translational modifications are important mechanisms that can support changes in regulatory interaction strength. The experimental evidence and implications to genetic regulatory interactions of these processes are summarized, for example, in a recent review [6]. These mechanisms confer gene regulatory interactions with the capacity to be modified and modulated; this is the substrate required for the random walk. Such variations are not only well-documented, they have in fact been directly implicated in coping with novel conditions. Examples include the case of cellular reprogramming [7,8] and adaptation to artificial gene rewiring [9,10].

We believe this biological evidence was not emphasized enough in the paper. This was modified in the revised version, with the appropriate references cited.

Reviewer #3

1. The authors exemplify their method on a rather general dynamic equation (1), with the main restriction being that the interaction function $\phi(x)$ asymptotically saturates for large x . While this is a rather common assumption in the context of gene-regulation, as the authors clearly note and appropriately cite, one still wishes to better understand the implications of such restriction. What would happen in case we implement non-saturating interaction dynamics? Would that harm the reported findings? It seems to me that saturating vs. divergent dynamics represents an essential distinction, with likely significant implications on the system's behavior, which may potentially enrich the picture currently portrayed.

This question is partly answered by our reply to Reviewer #1, where different versions of the nonlinear equations were examined (new Supplementary Sections 2.1, 2.2). The complete lack of a saturating function results in divergent dynamics of the equations of motion and are therefore not included in the analysis.

2. On that note, seems that the main motivation in the presented analysis relates to gene regulation, which follows the combined scale-free/bounded degree distribution, and likely the saturating $\phi(x)$. Cellular function, however is also driven by protein interactions - an un-directed scale-free biochemical interaction network with potentially non-saturating interaction terms. I do not feel that the paper needs a full-scale analysis of such interactions, it is rather rich and comprehensive in its current form, but, the authors may consider addressing this issue briefly, to better put their results in context.

We have addressed this point as suggested in the revised manuscript.

3. In Eq. (3) it seems that the phenotype $y(t)$ is time dependent, namely that the function to which the cell should evolve is not just a point in "phenotype-space", but a complete temporal function. Upon first reading I could not fathom how a system could ever successfully adapt to such a requirement, seeking a target network that not only satisfies a desired fixed point, but also features a specific temporal pattern. Reading on, I realized, that this was not the authors' intention, and indeed, the adaptation leads the system to a desired fixed point. This point should be made clear in the paper, and especially in the presentation of Eq. (3).

This was clarified and better explained in the revised manuscript.

4. The exploration follows a form of diffusion, depending on the square-root of the discrepancy from the desired phenotype. How important is this assumption to the convergence? The other arbitrary dependence is on the value of D , which the authors analyse comprehensively later on.

This is not a central assumption but a matter of convention. We have chosen the square root to adhere to common formulation of random walk. The important ingredient lies in the dependence of the function $M(y-y^*)$ which is highly nonlinear and discussed in more detail in Supplementary Section 1.4. We have made this distinction and the reference to this section more clearly in the revised text.

5. The one major issue that begs for additional insight is the way in which the degree distribution affects the adaptation process. The finding that a combined scale-free out-degree and bounded in-degree increases the probability of successful adaptation is central to this paper, and truly important. Hence rather than an intriguing empirical fact, the reader expects to gain more insight into the microscopic/mechanistic exchanges that enable this. This can probably be done through extensive numerical analysis, tracking the specific exploration steps that led to successful adaptation, and observing their relationship to the hubs. Even in the absence of an analytical explanation, I am certain, that such an analysis will also, along the way, shed light on the presented adaptation model and expose new facets and directions for researching this problem.

This is closely related to remark 5 of Reviewer #1. As detailed above, we now include a qualitative explanation based on the differential ability of hubs to coordinate changes in the state of nodes with lower out-degrees. This can explain the existence of attracting states in large networks (Fig. 5) and supports the adaptation of networks in which one or more of these attractors is compatible with the new demand. Please see detailed response above to remark 5 of Reviewer 1.

6. Minor corrections: Two citations misplaced (3,19) below Eq. (4); Page 3: "...substantially more abundant than" - should be that.

Corrected.

To summarize, I believe this is a very good paper, that can be further improved through few "adaptations", if the authors expand on the points above.

We thank this reviewer for his/her highly positive review and remarks.

References

- [1] R. Pinho, E. Borenstein, and M.W. Feldman. "Most networks in Wagner's model are cycling". *PloS one*, **7**, e34285 (2012).
- [2] H. Sompolinsky, A. Crisanti, and H.-J. Sommers. "Chaos in random neural networks". *Physical Review Letters*, **61**, 259 (1988).
- [3] R. Pinho, V. Garcia, M. Irimia, and M.W. Feldman. "Stability depends on positive autoregulation in Boolean gene regulatory networks". *PLoS Comput Biol*, **10**, e1003916 (2014).
- [4] A.S. Konagurthu and A.M. Lesk. "On the origin of distribution patterns of motifs in biological networks." *BMC Systems Biology* **2**, 73 (2008).
- [5] O.X. Cordero and P. Hogeweg. "Feed-forward loop circuits as a side effect of genome evolution." *Molecular biology and evolution* 23(10), 1931 (2006).
- [6] K.J. Niklas, S.E. Bondos, A.K. Dunker and S.A. Newman, "Rethinking gene regulatory networks in light of alternative splicing, intrinsically disordered protein domains, and post-translational modifications". *Front. Cell Develop. Biol.* **3**,8 (2015) .
- [7] J. Hanna, K. Saha, B. Pando, J. Van Zon, C.J. Lengner, ... & R. Jaenisch, "Direct cell reprogramming is a stochastic process amenable to acceleration". *Nature* **462**, 595-601, (2009).
- [8] Y. Buganim, D.A. Faddah, A.W. Cheng, E. Itskovich, S. Markoulaki, K. Ganz, ... & R. Jaenisch. "Single-cell expression analyses during cellular reprogramming reveal an early stochastic and a late hierarchic phase". *Cell*, **150**, 1209-1222 (2012).
- [9] S. Stern, T. Dror, E. Stolovicki, N. Brenner and E. Braun, "Genome-wide transcriptional plasticity underlies cellular adaptation to novel challenge". *Molecular Systems Biology*, **3**, 106, (2007).
- [10] Y. Katzir, E. Stolovicki, S. Stern and E. Braun, "Cellular plasticity enables adaptation to unforeseen cell-cycle rewiring challenges". *PloS one*, **7**, e45184 (2012).

REVIEWERS' COMMENTS:

Reviewer #1 (Remarks to the Author):

The authors have addressed all my previous comments in a satisfactory fashion. I have no further comments on the revised version.

I highly recommend this paper for publication in Nature Communications. I believe this work could trigger a burst of research activities in this area.

Reviewer #2 (Remarks to the Author):

In this revised version, the authors responded well not only to my comments, but also to the comments of two other reviewers that made pertinent points. They added a significant amount of new material that addresses concerns about the generality of the findings, and also put the work into more of a biological context. With these revisions, I can advocate publication.

Reviewer #3 (Remarks to the Author):

I am glad to see that the authors have addressed my comments in a sufficient manner. At this stage I recommend publication in Nature Communications.